## [Peer Review file · Nature]

Clonal-aggregative multicellularity tuned by salinity in a choanoflagellate

Corresponding Author: Dr Thibaut Brunet

Version 1:

Reviewer comments:

Referee #1

(Remarks to the Author)

The manuscript titled “Clonal-aggregative multicellularity entrained by salinity in one of the closest relatives of animals” by Ros-Rocher et al. highlights a new form of simple multicellularity in *Choanoceca flexa*, which combines clonal and aggregative behaviour. The authors also investigate the ecological context behind this mixed form of multicellularity in natural and experimental conditions. The genome assembly and SNP-calling give insights into the nature of aggregative multicellularity (kin recognition). Together, this manuscript explores the natural history of *C. flexa* and the potential adaptive logic of mixed clonal-aggregative multicellularity.

Major comments

The manuscript addresses an interesting and timely topic—the emergence and evolution of multicellularity in eukaryotes and offers valuable insights into the biology and natural history of choanoflagellates. However, I am not convinced it presents a significant conceptually important advance, and falls short in providing sufficient evidence to support several claims. My arguments follow below.

Claim 1. “*C. flexa* develops into motile and contractile monolayers of cells (or “sheets”) through a mixed, plastic mechanism: purely clonal processes, purely aggregative processes, or a combination of both”.

Clonal, simple multicellularity in *C. flexa* has been previously demonstrated, the hallmark being inverting curvature and switching between feeding and swimming ‘states’. While these behaviors are thought to represent a rudimentary form of coordinated multicellularity, the lab experiments described in this study do not, in my view, sufficiently support the claim that the “aggregative colonies” are bona fide multicellular and capable of coordinated behavior. For example, the claim that “aggregation was sufficient for *C. flexa* multicellular development” (e.g. lines 126–127) is not sufficiently substantiated. Specifically, it would be critical to demonstrate that aggregates exhibit coordinated multicellular behaviors akin to clonal colonies, such as curvature inversion and state switching. Perhaps I missed this result? The supplemental 3D images do not convincingly show a coordinated multicellular structure but rather appear as random clumps of cells without discernible patterns or behaviors. This observation could simply result from experimental conditions where cells are brought into proximity on a plate. The authors would need to demonstrate that aggregated cells (sheets arising from aggregates from two or more genotypes) exhibit coordinated behaviors, such as switching from feeding to swimming states. Furthermore, they must establish that this aggregative/multicellular behavior occurs in natural environments and is not merely a laboratory ‘artifact’. This would involve collecting “sheets” directly from field samples, manually isolating each of them and demonstrating that (i) individual sheets contain multiple genotypes and (ii) these potentially ‘aggregative’ sheets exhibit coordinated multicellular behavior.

Claim 2. “Multicellular development is controlled by salinity”

I am not sure that the authors showed that multicellular development is “controlled” by salinity? – I would say, more humbly, that in harsh conditions (such as high salinity) they form cysts. I feel that the interpretation of the salinity experiments involves a great deal of speculation. And crucially, I am not sure I saw evidence demonstrating that the ‘mixed strategy’ of multicellularity confers any selective advantage. The authors propose that “clonal-aggregative development allows fast and

reversible transitions between unicellular and multicellular lifestyles in this rapidly fluctuating environment.” But have they showed this? – as far as I see it, they do show that in harsh conditions there is encystation and that in lower salinity the organism form multicellular colonies that feed better on bacteria. Does't this show So as far as I can tell they show that there is an advantage in encystation versus multicellular forms (regardless of aggregative or clonal). In the manuscript they test a feeding advantage of the multicellular colonies, compared with encystation (in high salinity conditions) but what this actually means is that encystation is an advantage in stressful conditions (which has been shown previously in other systems, e.g. <https://www.nature.com/articles/s41598-020-75194-3>). The authors would need to show that each form of multicellularity (aggregative, clonal or mixed) results in a measurable fitness increase under the respective environmental conditions.

Section “Occurrence of multicellular *C. flexa* in the field is limited by salinity”

Lines 161-174 about sheets being associated with lower salinity: I looked closely at the supplemental excel data sheet (Supplemental data 492354_1_data_set_4557578_skmmm1.xlsx) and I am not sure I understand why the data from line 158-160 in the excel sheet exped-A has been excluded from the plots. The data show that sheets are also found in very high salinity (so they contradict the author’s conclusions). Did I miss something? Why were these data not included in the analysis? This means that the whole section on sheets being associated with lower salinity is incorrectly interpreted (sentences in line 162 and in line 171).

Sections “Multicellular development is entrained by evaporation-refilling cycles in nature”, “Multicellular development is entrained by evaporation-refilling cycles in the lab” and “*C. flexa* sheets dissociate and differentiate into unicellular cysts at high salinity”

I believe the authors expend considerable effort to explain what is essentially a trivial observation: encystation in response to salinity stress (as the authors themselves state in line 245). These sections and imminently descriptive and I am not sure I understand what is the novel concept and how that links to the mixed aggregative-clonal multicellularity.

Note that the idea that choanoflagellate cells can differentiate into various morphotypes (including multicellular forms) in response to changing aquatic environments, potentially providing a selective advantage, has already been explored (doi: 10.1002/jez.b.22941). Therefore, I am unclear on what novel concept is being presented here.

Line 157 “In 10 out of 79 splash pools” – what about the 69 other pools?

Line 166 “We found sheets in 14 out of 71 splash pools” - what about the 57 other pools?

Claim 3. “Different splash pools contain different strains of *C. flexa* between which aggregation is constrained by kin recognition, a hallmark of aggregative multicellularity”

Another point concerns the analysis of kin recognition. First, the kin recognition index is difficult to interpret, given the small apparent dynamic range in Fig 5H and the distributions in Fig S15B having a ‘normalised percentage’ y-axis ranging from -1 to 1. A more intuitive measure may be the (mean) absolute difference (in percentages) or other measures that preserve the percentage (or ratio) values. Second, in Fig S15C, the kin-recognition index should not have a negative value on the y-axis, since variance cannot have a negative value.

Note that the entire section needs to be proofread. Fig 5K is not described in the legend, Fig 5A is not cited in the text. In L342, it is unclear if the authors are meant to refer to S15-S16 (rather than S5-S6). Fig S15 legend refers to the wrong figures (L1177).

This leaves us with an open question: is there a link between kin recognition and SNPs on the transmembrane proteins? Perhaps the SNPs on these transmembrane proteins can be correlated in the PCA space?

Claim 4. “Challenging former generalizations about the choanoflagellate-animal lineage and expanding the option space for the development and evolution of multicellularity”

Given the phylogenetic position of *C. flexa*, the implication of mixed clonal-aggregative multicellularity (or L435 “evolutionary interconversions”) for metazoan evolution is very speculative. For example, in Fig S6D, the mixed clonal-aggregative multicellularity observed in *C. flexa* is likely not ancestral within choanoflagellates. Thus there is caveat that aggregative multicellularity may not have been observed thus far in other choanoflagellates. It would have been worth performing an ancestral state reconstruction on the natures of multicellularity (clonal, aggregative or mixed) and adapting the discussion to the inferred ancestral states. Currently, the reader is left wondering whether the animal ancestor may have also had a mixed form of multicellularity.

Other points

Regarding replication, the manuscript does not provide sufficient details or data to validate the robustness of the findings. The authors state that “all experiments were replicated and quantified at least twice” and that additional unquantified replications were performed to ensure repeatability. However, this level of replication sounds insufficient, and it is unclear what was repeated and how many times. The data from replicate experiments, including the number of independent observations (number of colonies observed, number of independent samples from independent pools), should be included. All experiments should be independently replicated at least three times, with clear data supporting the conclusions drawn.

Note that the use of the term “development” throughout the manuscript is not fully justified. First, there is no “development” in these organisms (in other words, there is no ‘developmental program’). Secondly, it is critical to demonstrate that what is referred to as “aggregative colonies” reflects true multicellular (albeit simple) behavior and is not the result of random cell proximity, as explained above.

The difference between the terms agglomeration and aggregation is unclear, did I understand correctly that cells aggregate

into sheets and then sheets can get together to form agglomerates? Is this really meaningful?? It is still unclear to me what is the 'unit organism', the sheets (aggregates)? or the agglomerates?

Improved visualisation. Fig S16: the values on the heatmap are not easy to see. Perhaps, the values can be rounded to 2 decimal points. Fig 5D: bootstrap values can be written directly. Fig S9D: images are pixellated. Fig 5E and 5F: scale bar needed.

Fig 2F: a control without DMSO treatment is missing. Each replicate begin with same number of cells.

L120: a space after the full stop is missing. L396: space needed "aggregativemulticellular".

Movie S4 – "Dissociated single cells reform sheets by cellular aggregation" I don't understand what I am seeing, is this a time lapse? Looking at the individual images I have a lot of trouble in understanding what is going on. The trajectory of each cell should have been tracked independently to be able to follow their pathway and clearly show what it is supposed to show (sheet reformation?). All the other 'movies' are 3D representations of groups of cells –are these 'multicellular units' ? they seem to be rather clumps of cells (see my comment above).

Which strain is used for the final genome assembly?

Single-sheet-bottlenecked culture vs clones. Unclear 344-346

L340: What is a high confidence SNP?

Supp. L987-988: the R version might be more informative than the RStudio version.

Control for bacterial density (w.r.t. salinity)

Aggregation vs agglomeration: more information.

No Statistics are presented for 2F.

Non-kin aggregates: are there difference in sheet size?

(Remarks on code availability)

Referee #2

(Remarks to the Author)

General comments:

This is a beautiful work bridging environmental biology with modern cell and developmental biology. It combines a series of meticulously designed lab and field experiments, including advanced live-imaging, fixed-imaging microscopy, isolation, sequencing, and genomic analysis of the choanoflagellate *Choanoeca flexa*, a close relative of animals. The authors make several key discoveries: they first show, using live-imaging, that multicellular sheets of *C. flexa* form not only through clonal division but also through aggregation—a previously unreported mechanism for choanoflagellates. Field experiments, along with lab experiments that replicated natural environmental conditions, reveal that the life cycle of *C. flexa* is driven by salinity fluctuations in its splash pool habitat. As salinity rises due to evaporation, multicellular sheets dissociate into cyst-like forms; when salinity decreases, multicellularity is restored via both aggregation and clonal division. This suggests an adaptive strategy for rapid multicellular development, likely to enhance bacterial prey capture during the short-lived splash pool conditions. Additionally, the authors provide a new, fully annotated genome assembly for *C. flexa* and uncover evidence that aggregation is constrained by kin recognition, reducing potential conflicts. This elegant and thorough work deepens our understanding of the flexible, environment-driven mechanisms that may have shaped the evolution of multicellularity. In principle, this is a beautifully executed study with precise experiments. However, I offer a few points below that could help strengthen the authors' interpretations and conclusions.

Major points :

1. Structural flow: The current structure of the paper feels like two distinct stories: one focused on the aggregation process and another centred on the life cycle of *C. flexa* in its dynamic splash pool environment. While each component is well-explored, the connection between them is limited, which makes the overall narrative somewhat less cohesive. A potential approach could be to reframe the story around the entire life cycle of *C. flexa*. This would start by describing the environmental dynamics of splash pools, emphasizing the role of salinity fluctuations in driving the organism's life stages, from desiccation and cyst formation to the rapid re-emergence of multicellularity. By highlighting that multicellularity reappears faster than clonal division alone could account for, the narrative could naturally introduce the role of aggregation as a key mechanism for rapid colony formation. This structure would make it easier to integrate the data on aggregation and kin recognition, showcasing them as adaptive responses shaped by the fluctuating environment. Alternatively, if the current structure is maintained, it would be beneficial to strengthen the textual links between the environmental life cycle data and the multicellular behavior. Explicitly connecting the changes in salinity to the shifts in multicellularity throughout the text would provide a more cohesive and integrated interpretation of the findings. To better

bridge the two parts of the study and directly link the life cycle dynamics with multicellular behavior, I recommend additional experiments focused on the speed of sheet development under different stress conditions: Measure the speed of multicellular sheet formation in cultures that have been pre-stressed by high salinity versus those that have not been salinity-stressed. This would involve dissociating all cultures into single cells, with one group exposed to prior salt stress and the other group left unstressed.

Assess the rate of sheet formation both in isolated (stressed or unstressed) and mixed conditions (stressed cells combined with unstressed cells) after switching to conditions favouring division and sheet formation. Time-lapse imaging could be used to quantify the speed of colony formation in each case. In parallel, measure the cell division rates under these conditions over time (as some division rate data may already be present in the study) to directly compare the contribution of clonal division versus aggregation in sheet formation. These experiments would help clarify the link between the life cycle, particularly salinity stress, and the "driving" force behind multicellular sheet formation.

I know some experiments are already within the paper, they however seem scattered and somewhat not cohesively presented.

2. Aggregation and orientation:

The authors present compelling evidence for aggregation through several movies and figures, showing cell aggregation at the periphery of developing sheets. They strengthen this claim with experiments using the division inhibitor aphidicolin, demonstrating that aggregation can occur independently of cell division. Interestingly, the data suggest that aggregation initially happens in a disordered manner, followed by a reorientation phase where the cells align consistently. Additionally, the use of three genetically distinct *C. flexa* strains highlights the role of kin recognition in aggregation, suggesting a non-random, selective process. While the data are convincing, I still have several questions and suggestions for further exploration:

- Could part of the observed aggregation be random, where cells become "entrapped" together initially, only to divide and reorient later? This is an open question (and one I'm unsure about or whether there is a clear way to showcase it). To explore this, one experiment could involve adding dissociated, fluorescently labeled single cells of one color onto pre-formed colonies of a different color. Comparing this scenario (few colonies with many single cells) to a control where only single cells of distinct colors are mixed could help differentiate random aggregation from more directed, kin-based aggregation?
- The collar complex likely plays a central role in the aggregation process. A potential experiment could involve testing the aggregation behavior of single cells that lack a functional collar complex. This could be achieved by pre-treating cells with actin inhibitors or exposing them to high salinity (as suggested earlier), then mixing these non-collar cells with pre-formed colonies of a distinct color. Monitoring whether aggregation occurs and in what proportion could provide insights into the role of the collar complex in the initial cell-cell adhesion process.
- Just thinking out loud here, but would you expect that aggregation per se has different properties than clonal division? Would an experiment where cells only formed through aggregation (Aphidicolin treated) can be subjected to precise small sonication, vs clonal-aggregative together? Basically, trying to see whether there maybe another advantage here than speed? but some sort of biomechanical reinforcement? (again, this maybe beyond the scope).
- The reorientation of cells within the sheet is a particularly interesting phenomenon. It raises the question: in the aphidicolin experiments (where cell division is inhibited), are the aggregated cells well oriented or not? If the cells achieve proper orientation even without division, this suggests that reorientation is driven by mechanisms independent of cell division. If not, it would imply that while aggregation can occur without division, reorientation may still depend on subsequent divisions. It would be helpful if the authors included 3D images of the sheets from the aphidicolin experiments (I checked the supplementary materials but did not find detailed imaging of cell orientation in these conditions). Moreover, in the case where cells do not fully re-orient when only forming aggregate, it would be interesting to assess the flows ? and or efficiency of feeding ? (but I understand it maybe beyond the scope of this study).
- I would encourage the authors to discuss these observations as two distinct but connected processes: (1) initial aggregation and (2) subsequent cell reorientation. It would be valuable to speculate on the possible signaling mechanisms involved in each stage. Given the known importance of calcium signaling, an experiment using EDTA/EGTA (to chelate calcium ions) could help determine whether aggregation and/or reorientation are dependent on calcium-mediated interactions for instance (although I can understand that it may be beyond the scope of this study). I just think it would be very interesting to find a condition where cells aggregate, but are unable to fully re-orient. The authors use three distinct *C. flexa* strains to demonstrate that aggregation may be constrained by kin recognition, limiting aggregation to genetically similar cells. While this is a valuable finding, it would be strengthened by the inclusion of an outgroup, such as *S. rosetta*. Adding *S. rosetta* cells to the aggregation assay could help determine whether aggregation is driven purely by kin recognition or if some degree of random entrainment occurs. If *S. rosetta* cells do not aggregate with *C. flexa*, it would support the idea that kin recognition plays a crucial role. Conversely, if they become entrapped, it might indicate a more passive, non-selective aggregation mechanism at play.

Minor points:

- This may be a personal perspective, but I find that the beauty of this study lies in its standalone contribution, without necessarily invoking the evolutionary history of animal multicellularity. The authors have done an excellent job throughout most of the manuscript in avoiding overemphasis on the origins question, keeping the focus on the unique ecological and cellular biology aspects of *C. flexa*. However, I did notice one instance in the introduction where this theme emerges: "As in animals, multicellular development is clonal in all choanoflagellate species investigated thus far, which contrasts with the occurrence of aggregative multicellularity in more distantly related lineages, such as filastereans and dictyostelid amoebae. These observations, together with the clonal nature of animal embryogenesis, have inspired the hypothesis that animals evolved from organisms with a simple form of clonal multicellularity, akin to that observed in certain modern choanoflagellates and in other close relatives of animals such as ichthyosporeans."

While this statement is accurate, it indirectly suggests that the authors' discovery of clonal-aggregative multicellularity might challenge the hypothesis that animal ancestors had only a simple clonal multicellular organization. The results of this study,

in my view, imply that the observed clonal-aggregative behavior in *C. flexa* may be an adaptation specific to its environmental context, rather than a broader ancestral trait of choanoflagellates. The absence of such behavior in other choanoflagellate species to date suggests it might be a specialized adaptation, evolved in response to environmental pressures rather than a retained ancestral capability. However, given the prevalence of aggregation across various protist lineages, we cannot completely rule out the possibility that early ancestors had both clonal and aggregative capacities. I realize that this is a nuanced and inconclusive point, and the true answer likely lies somewhere in between. The way the introductory sentences are framed either necessitates a more thorough discussion of both scenarios later on or could benefit from a slight rephrasing to downplay the evolutionary question upfront. This would allow the focus to remain on the unique eco-evo-cell biology story of *C. flexa*, which in itself is a remarkable narrative. Of course, I acknowledge that this is a subjective suggestion.

Figure comments:

- In general, figures are very beautiful and well built. One comment would be to normalize the colour palettes for the collar. In some images its in blue, in others in green and in sketches it's in green. I understand that sometimes phalloidin is used in the 405 and others in the 488 but I think normalizing these across the videos and figures (maybe grey ?) would help readers to better follow through without changing anything of the narrative.
- Figure 5, I wonder whether the red and green in J drawings of the domain are colour-blind compatible?

(Remarks on code availability)

Not my expertise

Referee #3

(Remarks to the Author)

This elegant paper from the Brunet lab reveals an unexpected mode of multicellular development in *Choanoeca flexa*, one of the closest living relatives of animals. The authors demonstrate that *C. flexa* can form multicellular sheets through both clonal division and cellular aggregation, and show this developmental plasticity is environmentally regulated by natural salinity cycles in splash pools, the organism's native habitat.

The authors establish that *C. flexa* can form multicellular sheets through pure clonal development (cell division with retained adhesion), pure aggregation of individual cells, or a combination of both mechanisms. In its natural splash pool habitat, *C. flexa* alternates between multicellular sheets at lower salinities and unicellular cysts at high salinities or during desiccation, a cycle driven by natural evaporation-refilling dynamics. Importantly, the multicellular form shows enhanced bacterial prey capture compared to single cells, providing a compelling argument for the benefits of multicellularity (this is in addition to the morphological transitions between motile 'cup' shaped colonies that flatten into a feeding mode once they are in the light, which I hypothesize also allows them to better find prey). Different splash pool populations show genetic divergence and kin recognition during aggregation, with polymorphic cell surface proteins potentially mediating strain recognition.

This is an exceptionally rich, complete story. The technical execution of this work is outstanding. The authors employ an impressive array of complementary approaches, including field ecology and environmental monitoring, time-lapse microscopy and cellular imaging, genomics and phylogenetic analysis, behavioral assays, and quantitative analysis of cellular interactions. The scholarship is exceptional, with the work thoughtfully positioned within the broader context of multicellularity evolution, development, and ecology. The authors carefully build their argument through a logical progression of experiments, each providing multiple lines of evidence for their conclusions. This is, simply put, an stand-out paper. The integration of lab and field approaches is especially noteworthy, given how hard it can be to understand the context in which the behaviors of extant organisms likely evolved.

I enthusiastically support publication of this paper, largely as is. I have few suggestions for improvement, and my suggestions are truly minor. The authors should not feel obligated to do any of these things simply to appease me: only do them if you think they make the paper better.

First, the "kin recognition index" could be more intuitively presented using established methods for measuring preferential assortment. The segregation index described by Estrela and Brown (2013, PLoS Computational Biology) provides a more interpretable measure, scaled between -1 and 1, where 0 represents random mixing and 1 represents pure kin recognition. This would be more accessible to most readers than the current variance-based approach, where contextualizing the measurements is challenging. It's also a more useful metric, as assortment is equivalent to 'relatedness' in inclusive fitness calculations, as it's a weighted mean scalar of preferential interactions among clonemates. It's a nice way to analyze this data.

Second, the paper's framing of clonal development and aggregation as mutually exclusive developmental modes might be a touch too strong. While the authors are right that these are often *presented* as a dichotomy, mixed strategies are fairly common - yeast can flocculate while also forming clonal groups, *Chlamydomonas* does both, some sponges, plants and fungi can fuse while growing clonally, bacterial biofilms often involve mixed strategies of clonal and aggregative development, etc. What's particularly interesting is not that *C. flexa* is the first organism shown to do both, but a) the mechanisms seem to be more tightly regulated as part of their life cycle than what you see in the examples I cited above, and b) this opens the door to aggregation playing a role in the origin of animals, which nobody has really spent much time

thinking about.

Third, the adaptive significance of kin recognition warrants deeper exploration beyond the standard explanation of cheater prevention. I personally think people worry too much about the threat posed by cheats. In many simple, early multicellular organisms, cheating may not be a major selective pressure. If there isn't an obvious common good within *C. flexa* colonies that would tempt cheaters, alternative explanations for kin recognition should be considered. One compelling possibility is that the benefit lies in generating genetically-structured groups - not to prevent evolutionary deterioration through cheating, but to enable evolutionary construction and the origin of new multicellular innovations.

This works through a fundamental mechanism: in the absence of assortment, selection acting on the traits of groups does not result in evolutionary change (no change in allele frequencies over generations). However, when genotypes are positively assorted within groups, as they are in this case due to kin recognition, there exists a covariance between the genotypes of individuals within groups and selection acting on group traits. At an extreme, when groups are clonal, then selection acting on the traits of groups is acting on the genes within the cells of those groups that created the group-level trait under selection. While clonality is obviously the best for this, any positive assortment creates a statistical linkage between the traits of groups and the fitness of alleles within the group that underpin the expression of multicellular traits.

C. flexa has evolved sophisticated multicellular behaviors (i.e., cells responding to light by changing their collar angle, that changes the shape of the colony from a flat sheet to a motile cup) where there is a clear linkage between the traits of cells and the traits of groups. Selection acting on the behaviors of groups would be much more efficacious if those groups were highly assorted, as opposed to random grabs of all the different genotypes that exist in a splash pool. Thus, kin recognition might be more about enabling the evolution of coordinated multicellular behaviors than preventing exploitation. If this argument does not make sense to you, no worries, reach out to Dr. Gee and ask him for my name. He has my permission to waive anonymity, and we can chat about it. I think a quick zoom session with a virtual white board would make it super clear.

Bottom line: this is a truly beautiful, elegant paper, one of the best yet from a rising star in the field of multicellularity. It's a complete story leveraging many diverse tools of modern biology that should leave everyone happy- if you're a field biologist, you'll be envious of the genomic/lab data that can be brought to bear, and if you're a lab scientist, you'll be envious of such a clean field component that provides so much power for understanding the historical context of selection.

(Remarks on code availability)

Version 3:

Reviewer comments:

Referee #1

(Remarks to the Author)

The authors have made an excellent job in answering the comments, I have no further concerns, congratulations for the very nice work.

(Remarks on code availability)

Referee #2

(Remarks to the Author)

The authors have done an exceptional job addressing all my previous comments, and the paper is now far stronger than before. They've clearly put in a huge amount of work (actually quite a spectacular amount of work), carefully tackling every point raised, including some that were, frankly, unreasonable from Reviewer 1.

This is a game changing piece of work for how we think about choanoflagellate multicellularity. The data quality, depth, and clarity of the story are remarkable. I also want to emphasize that the request for an additional replicate, one that would have required another field trip to Curaçao, was unnecessary and unrealistic, and I fully support the authors decision not to pursue it. With the new data testing only aggregative sheets vs clonal ones, with all the new image datasets, sequencing, quantifications, and just a new written manuscript, this has over exceeded my expectation for a revision.

Overall, this is a truly important and field defining study. I'm fully and enthusiastically in favor of publication in its current form, and would hardly understand how such work can't make it through.

(Remarks on code availability)

Referee #3

(Remarks to the Author)

I am very happy with these revisions (both for my own comments and those of the other referees). I'd like to restate that I think that this paper is very deserving of being published in Nature: not only are the conceptual issues critical for understanding animal origins (i.e., understanding the life cycles of animal relatives, which may inform our understanding of animal ancestors, is critical for understanding how simple groups of cells evolved into functionally-integrated organisms), but the experiments and scholarship are top notch. Vanishingly few papers combine fieldwork and field-leading cell biology, and knowing the environment these organisms exist in, and thus likely evolved in, makes the cellular mechanisms all the more powerful. Fun fact: *S. rosetta*, the choanoflagellate that Nicole King brought to fame, has been isolated from the ocean exactly one time, and has never been seen again. Essentially all evolutionary hypothesis about this organisms remain ungrounded in ecological reality, and yet it's still been profoundly impactful for our understanding of animal multicellularity. The Brunet lab is doing some of the most elegant work in the field- with this paper being among their finest work to date.

(Remarks on code availability)

We sincerely thank all reviewers for their constructive and thoughtful comments, which have significantly improved the quality of our paper. We greatly appreciate your time, energy, and expertise, and over the past months, revising our study in the light of your feedback has been our top priority. The revised paper is now substantially stronger and more convincing as a result. Thank you.

To briefly summarize the main changes:

- We have now established experimental conditions under which colonies form purely by aggregation. We have used this to show that colonies formed purely by aggregation are equivalent in morphology and behaviour to control colonies formed under permissive conditions for cell proliferation. We support this point with multiple and extensive morphometric and behavioural quantifications.
- We have performed time-resolved morphometric quantifications during the aggregation process, as well as high-resolution time lapse movies with cell tracking. These show that aggregation is a multi-step process: an early phase of random collision and adhesion in variable geometries gives rise to irregular clumps, followed by cell re-orientation that supports maturation into geometrically regular and polarized cell monolayers. This re-orientation process is independent of cell division, consistent with the fact that purely aggregative sheets have regular morphology and behaviour.
- We show that aggregation is an active and specific process, as it can only be performed by living cells, and as *C. flexa* does not aggregate with choanoflagellates of another species (*S. rosetta*).
- We show that aggregation and colony integrity require a microvillar collar scaffolded by F-actin, further substantiating a direct link between encystation (with associated collar retraction) and loss of multicellularity.
- We have developed a pipeline to quantify the relative contribution of clonality and aggregation to colony growth. This allowed us to show how the balance between both processes is modulated by environmental conditions. The resulting dataset reinforces our interpretation of clonal-aggregative multicellularity as a versatile strategy for the robust re-establishment of colonies across a broad range of environmentally relevant conditions.
- We have revised our analysis of high-polymorphism genes by using improved pipelines that now consider indels and not just SNPs. Without drastically affecting the main message, our improved analysis now recovers cadherin-like domains as the most enriched domain among genes bearing signature of diversifying selection.
- We have used a new quantification of kin recognition – the ‘segregation index’, suggested by Reviewer 1 and Reviewer 3 – directly based on the relative proportions of each genotype.
- We have toned down the discussion of the risk of cheating as a potential selective pressure for kin recognition and instead emphasized the co-evolution of genotypes for the coordination of complex behaviours.
- Finally, we have clarified our main message: our case study of *C. flexa* shows that choanoflagellate multicellularity is much more diverse than anyone in the field expected. This forces a revision of prevailing ‘textbook’ generalities about choanoflagellates,

illustrates the importance of characterizing organisms in their natural environment, and establishes *C. flexa* as a new model for eco-evo-devo.

Referees' comments:

Referee #1 (Remarks to the Author):

The manuscript titled “Clonal-aggregative multicellularity entrained by salinity in one of the closest relatives of animals” by Ros-Rocher et al. highlights a new form of simple multicellularity in *Choanoceca flexa*, which combines clonal and aggregative behaviour. The authors also investigate the ecological context behind this mixed form of multicellularity in natural and experimental conditions. The genome assembly and SNP-calling give insights into the nature of aggregative multicellularity (kin recognition). Together, this manuscript explores the natural history of *C. flexa* and the potential adaptive logic of mixed clonal-aggregative multicellularity.

Major comments

The manuscript addresses an interesting and timely topic—the emergence and evolution of multicellularity in eukaryotes and offers valuable insights into the biology and natural history of choanoflagellates. However, I am not convinced it presents a significant conceptually important advance, and falls short in providing sufficient evidence to support several claims. My arguments follow below.

We thank you for stating that our manuscript offers valuable insights into choanoflagellate biology, and for your constructive and detailed feedback. We have performed extensive revision experiments to address all the points you raised. Additionally, we have rewritten our introduction and discussion sections to better convey the conceptual contribution of our study.

Claim 1. “*C. flexa* develops into motile and contractile monolayers of cells (or “sheets”) through a mixed, plastic mechanism: purely clonal processes, purely aggregative processes, or a combination of both”.

Clonal, simple multicellularity in *C. flexa* has been previously demonstrated,

Thank you for contextualizing our study within our earlier work on *C. flexa*. We would like to clarify an important distinction: clonal multicellularity has not been previously documented in *C. flexa*. While Brunet et al. (Science 2019) demonstrated that laboratory cultures could be established from a single cell, we did not determine whether the resulting colonies arose clonally (through cell division without sister-cell separation) or through aggregation of independent cells.

To illustrate this distinction, consider the social amoeba *Dictyostelium discoideum*: although laboratory cultures can be (and routinely are) established from a single cell, *D. discoideum* achieves multicellularity purely through aggregation: cells proliferate solitarily and only later aggregate into colonies as a distinct process.

The present study is the first to demonstrate clonal multicellularity in *C. flexa*, using live imaging of single cells forming clonal colonies (**Figure 1E and Movie S1**) and documenting the contribution of clonal division to colony expansion (**Figures 1G and S1, Movies S2-S3**).

the hallmark being inverting curvature and switching between feeding and swimming ‘states’. While these behaviors are thought to represent a rudimentary form of coordinated multicellularity, the lab experiments described in this study do not, in my view, sufficiently support the claim that the “aggregative colonies” are bona fide multicellular and capable of coordinated behavior. For example, the claim that “aggregation was sufficient for *C. flexa* multicellular development” (e.g. lines 126–127) is not sufficiently substantiated. Specifically, it would be critical to demonstrate that aggregates exhibit coordinated multicellular behaviors akin to clonal colonies, such as curvature inversion and state switching. Perhaps I missed this result?

We agree that the previous version of the manuscript lacked sufficient evidence demonstrating that sheets formed by aggregation were capable of normal collective behaviour. We have performed additional experiments to address this point:

- First, we have established a protocol to inhibit cell division without affecting aggregation by dissociating sheets and transferring single cells from 1X to 2X seawater salinity. This mild hypersaline shock does not cause encystation but robustly interrupts cell division for more than 24 hours, thus allowing us to overcome the toxicity associated with prolonged treatment using cell cycle inhibitors. We have now used this method to produce sheets purely by aggregation. The new protocol is detailed in the revised Materials and Methods.
- We then investigated the response of the resulting purely aggregative sheets to light-to-dark transitions (which induces collective inversion behaviour in *C. flexa*). Both aggregative and control sheets displayed robust collective behaviour. This is now documented by quantifications of decrease in sheet area under light-to-dark transitions (indicative of inversion) in our new **Figures 5G** and **S19**:

We have further supported this point with live imaging data showing that chimeric sheets – formed by aggregation of cells stained with two different colours under 2X salinity (and thus produced by ‘pure’ aggregation) – invert in response to light-to-dark transitions (see new **Movie S19**).

Finally, we also show that the efficiency of bacterial prey capture by purely aggregative sheets is comparable to that of control sheets, and that, in all cases, multicellular sheets present a significant feeding advantage over single cells (see new **Figure 5H**).

The supplemental 3D images do not convincingly show a coordinated multicellular structure but rather appear as random clumps of cells without discernible patterns or behaviors.

The reviewer raises an important point: are colonies formed purely by aggregation as morphologically regular as those observed in ordinary cultures, that result (at least in part) from cell proliferation?

To address this, we have now generated an extensive new dataset from both fixed and live samples. To interpret these data, one must first distinguish two phases of the aggregation process:

- Immediately after the initial encounters between single cells or separate sheets, aggregates usually have irregular morphologies. This presumably reflects random collisions between cells followed by adhesion in variable orientations. This is especially prevalent during the first 2 hours in our aggregation time course experiments (**Figures 2E and S2B, Movies S7-S8**) as well as at early time points in our new aggregation timelapse videos (**Figure 2C-D, Movies S5-S6**). Our images of early aggregates intentionally illustrate this initial, disordered stage of the process.
- Starting ~30 minutes after initial aggregation, aggregates progressively become more regular via cell reorientation, and eventually acquire the stereotypical morphology characteristic of sheet colonies: polarized monolayers of cells with regular curvature and all flagella oriented in the same direction. This later, organized stage is depicted by images of mature aggregates (**Figure 2E and S2C, Movies S9-S11**).

We now clarify these two phases in the text:

“Early aggregates were often irregular in shape, reflecting initial collision and adhesion in diverse random orientations. Aggregates later underwent morphological maturation by cellular rearrangements and reorientation into polarized monolayers with canonical C. flexa sheet morphology (Figure 2D; movie S6).”

As we agree that this point was not sufficiently evident in our earlier figures and required more extensive quantification, we have now acquired new Airyscan confocal stacks and developed morphometric analyses to quantitatively compare the morphology of aggregative and control sheets. The new images and corresponding figure panels now show more clearly the polarized monolayer architecture of aggregative *C. flexa* colonies, compared to controls (**Figure 5D-E**):

c-c: collar-collar contacts

We computed four morphometric parameters to quantitatively compare the morphology of control and aggregative sheets: (1) colony size (area and number of cells per colony); (2) angle between the collars of neighbouring cells (a measure of cell-cell alignment); (3) proportion of aligned cells (*i.e.*, of cells with apico-basal polarity aligned with their neighbours); (4) circularity of the colony outline.

α : collar-collar angle. Cells are defined as 'misaligned' if $\alpha > 120^\circ$.

These metrics show that the morphology of colonies resulting from aggregation is statistically indistinguishable from that of control colonies (**Figures 5F** and **S17H-J**):

Moreover, morphometric quantification across our earlier aggregation time course confirms the trend evident from the images: early aggregates display high geometric variability but progressively mature into regular, polarized sheets (**Figures 2G-G'** and **S2B**):

By the end of maturation, morphometric values converge towards values typical of control sheets (average collar-collar angle $\sim 40^{\circ}$, $\sim 100\%$ of cells properly aligned). Notably, morphological variance (as quantified by collar-collar angle variance) dramatically decreases during maturation (**Figure 2G**), illustrating progressive establishment of a regular sheet morphology.

This observation could simply result from experimental conditions where cells are brought into proximity on a plate. The authors would need to demonstrate that aggregated cells (sheets arising from aggregates from two or more genotypes) exhibit coordinated behaviors, such as switching from feeding to swimming states.

As detailed above, we have now demonstrated that aggregative sheets display collective inversion behaviour between flagella-out and flagella-in states in response to light-to-dark transitions (**Figures 5G** and **S19**; **Movie S19**). Additionally, we have now also demonstrated collective prey capture in aggregative sheets (see below). Taken together, our revised manuscript strongly supports the fact that aggregative colonies display both regulated morphology and collective behaviour.

Moreover, we want to emphasize that cells in our experiments are not merely “brought into proximity on a plate”: our aggregation assays were conducted at *C. flexa* cell densities equivalent to those measured in nature. Consequently, the density-dependent processes observed in our experiments are highly likely to be ecologically relevant under natural conditions.

Furthermore, they must establish that this aggregative/multicellular behavior occurs in natural environments and is not merely a laboratory ‘artifact’. This would involve collecting “sheets” directly from field samples, manually isolating each of them and demonstrating that (i) individual sheets contain multiple genotypes and (ii) these potentially ‘aggregative’ sheets exhibit coordinated multicellular behavior.

Here, the reviewer suggests investigating chimerism in colonies collected in the wild. This is a good idea and part of our planned future research. However, this would not, on its own, represent a conclusive test of aggregative multicellularity for the following reason: even in species that exhibit purely aggregative multicellularity, resulting colonies often contain a single genotype. This results from both habitat structure (which increases the likelihood that cells encounter close relatives) and kin recognition (which excludes non-relatives from aggregates). This has been shown in *Dictyostelium discoideum*, where ~80% of fruiting bodies contain a single genotype, despite forming purely by aggregation and comprising up to 2 million cells (PMID 17496139). Thus, investigating chimerism in wild-collected sheets would not directly test the reality or prevalence of aggregation but would instead quantify the extent of genotype dispersal within the

habitat and the strength of kin recognition. Moreover, such work would require a new field expedition to Curaçao and, to be performed at genomic scale, would require optimization of single-cell whole-genome sequencing for *C. flexa* – a technique that remains challenging even in well-established model systems and that is the topic of a 5-year collaborative grant we have recently submitted. For these reasons, this very interesting research direction lies beyond the scope of this study. We also want to point out that in *Dictyostelium*, more than 70 years separated the first description of aggregation (Raper 1935) from investigations of genetic chimerism in the wild (Gilbert et al. 2007). Aggregation was already well-established and extensively studied before the latter.

Instead, the relevance of aggregation to the life cycle of *C. flexa* is supported by the fact that our laboratory experiments showing aggregation extensively replicate the relevant environmental parameters of Curaçao splash pools, including *C. flexa* cell density (a key limiting factor for aggregative multicellularity), salinity, temperature, evaporation rate, and bacterial community (*H. oceanii* is native to Curaçao and was co-isolated with *C. flexa*). Moreover, aggregation gives rise to morphologically and behaviourally wild-type colonies and is an active, regulated process, as it requires live cells and is both strain- and species-specific. Together, these observations make aggregation highly likely to be relevant to the natural life history of *C. flexa*.

Claim 2. “Multicellular development is controlled by salinity”

I am not sure that the authors showed that multicellular development is “controlled” by salinity? – I would say, more humbly, that in harsh conditions (such as high salinity) they form cysts. I feel that the interpretation of the salinity experiments involves a great deal of speculation.

We have revised the title of this section, clarified the text, and performed additional experiments to support our conclusions. Our key point was that encystation inevitably causes the loss of multicellularity: indeed, when differentiating into cysts, cells retract the microvilli that connect them to neighbouring cells within colonies. We have now independently tested this point by treating colonies with latrunculin B, an actin polymerization inhibitor that causes fast microvilli retraction (independently of salinity or encystation). This treatment caused dissociation of sheets into single cells within minutes, further substantiating the mechanistic link between microvillar retraction (which occurs naturally during encystation under hypersaline stress) and multicellular-to-unicellular transition. We have added these data to the paper (**Figure S15A-C**).

To make sure this point is clear, we no longer refer solely to encystation, but rather to a joint process of “dissociation-encystation”, through which multicellularity is lost as cells encyst and subsequently regained when cells de-differentiate from cysts into flagellates with microvillous collars:

“This confirms that microvillar integrity is necessary for multicellularity and further supports the idea that sheets undergo a joint dissociation-encystation process in hypersaline conditions.”

Finally, we have toned down the statement “Multicellular development is controlled by salinity” to more accurately reflect our empirical observations. Our revised abstract now reads:

“C. flexa undergoes reversible transitions between unicellularity and multicellularity during cycles of evaporation and refilling”.

And crucially, I am not sure I saw evidence demonstrating that the ‘mixed strategy’ of multicellularity confers any selective advantage. The authors propose that “clonal-aggregative development allows fast and reversible transitions between unicellular and multicellular lifestyles in this rapidly fluctuating environment.” But have they showed this? – as far as I see it, they do show that in harsh conditions there is encystation and that in lower salinity the organism form multicellular colonies that feed better on bacteria. Doesn't this show So as far as I can tell they show that there is an advantage in encystation versus multicellular forms (regardless of aggregative or clonal). In the manuscript they test a feeding advantage of the multicellular colonies, compared with encystation (in high salinity conditions) but what this actually means is that encystation is an advantage in stressful conditions (which has been shown previously in other systems, e.g. <https://www.nature.com/articles/s41598-020-75194-3>). The authors would need to show that each form of multicellularity (aggregative, clonal or mixed) results in a measurable fitness increase under the respective environmental conditions.

Thanks for raising this point. We have now collected additional evidence showing a feeding advantage in both control and aggregative sheets. Specifically, we measured bacterial capture efficiency and found that both types of sheets exhibit a clear and comparable prey capture advantage over single cells (**Figure 5H**):

Furthermore, as detailed in our response to Reviewer 2, we have now shown that mixed clonal-aggregative multicellularity allows the re-establishment of multicellularity across a broader range of conditions than either clonality or aggregation alone. This further supports the idea that this mixed strategy confers a selective advantage.

Section “Occurrence of multicellular *C. flexa* in the field is limited by salinity” Lines 161-174 about sheets being associated with lower salinity: I looked closely at the supplemental excel data sheet (Supplemental data 492354_1_data_set_4557578_skmmm1.xlsx) and I am not sure I understand why the data from line 158-160 in the excel sheet exped-A has been excluded from the plots. The data show that

sheets are also found in very high salinity (so they contradict the author's conclusions). Did I miss something? Why were these data not included in the analysis? This means that the whole section on sheets being associated with lower salinity is incorrectly interpreted (sentences in line 162 and in line 171).

The reason these two data points were not plotted was stated in the supplementary data sheet: “One **sick** sheet found (omitted)” (emphasis added). Although this statement was concise and informal (as is typical for annotations of raw data collected in the field, which were included with the manuscript for full transparency), it nonetheless provided a clear reason. To dissipate any doubt on this point, we have now replaced this statement with a more detailed description of the phenotype we initially referred to as “sick” (Exped-A tab in **Table S1**):

“A single irregular, non-motile sheet was observed.”

We now mention these two observations in the main text:

*“In all cases, actively swimming and inverting *C. flexa* sheets were no longer observed after salinity crossed a 128 ppt threshold during gradual evaporation (Figures 3H and S8), consistent with results from Exped-A and Exped-B (Figures 3E-H and S8). **Interestingly, we observed two isolated colonies with an apparently stressed phenotype (irregular outlines, loose cell packing, no flagellar beating and no inversion behaviour) at 212 and 200 ppt in two different splash pools at Day 8 of sampling (Sp64 and Sp69; Figure S8E-F; Table S1). These irregular colonies were not observed at later time points, suggesting they had undergone death or dissociation (Figure S8F; Table S1). Similarly, evaporation in the laboratory of a natural splash pool sample containing sheets led to sheet disappearance (Figure S9). These observations confirmed that the multicellular form of *C. flexa* does not tolerate high salinity.**”*

Finally, to increase transparency in our representation of the data, we no longer omit these two samples from the relevant plots, but now depict them with a specific symbol: see the new **Figure 3E**:

Note that even if non-motile sheets were artificially conflated with active sheets, the difference in salinity between pools with and without sheets would remain highly significant ($p=8e-6$, compared to $p=1.6e-6$ if only active sheets are included in the analysis).

Non-motile sheets are also depicted in our revised time-course graphs (**Figures 3F and S8F**):

Dissociation-encystation is not an instantaneous process, and one should therefore expect to sometimes observe sheets in intermediate stages of dissociation in nature. The phenotype of non-motile sheets closely resembled that of colonies in the process of dissociation-encystation

during progressive evaporation in the laboratory (see **Figure 4D**, 112 ppt panel). Importantly, these dissociating sheets are clearly different from the active (motile and contractile) sheets observed at low salinity. Together, these two observations thus support, rather than contradict, our interpretation.

Sections “Multicellular development is entrained by evaporation-refilling cycles in nature”, “Multicellular development is entrained by evaporation-refilling cycles in the lab” and “*C. flexa* sheets dissociate and differentiate into unicellular cysts at high salinity” I believe the authors expend considerable effort to explain what is essentially a trivial observation: encystation in response to salinity stress (as the authors themselves state in line 245). These sections are imminently descriptive and I am not sure I understand what is the novel concept and how that links to the mixed aggregative-clonal multicellularity.

We agree that cyst ultrastructure is not the focus of the paper and have now accordingly moved the corresponding panels (original **Figure 4N-O**) to the supplementary material (now **Figures S12H** and **S14C**). As summarized above, encystation is relevant to this study only insofar as it necessarily entails loss of multicellularity, rather than as a cellular process in itself. To clarify our focus on multicellularity, our revised text now refers to “dissociation-encystation”.

Note that the idea that choanoflagellate cells can differentiate into various morphotypes (including multicellular forms) in response to changing aquatic environments, potentially providing a selective advantage, has already been explored (doi: 10.1002/jez.b.22941).

We thank the reviewer for citing this paper, which provides a good summary of the state of the art on choanoflagellates, and is a useful comparison point to show the novelty of our work. The cited study is a review focusing on *Salpingoeca rosetta*, the only other choanoflagellate species with a well-characterized life history. Our study significantly exceeds the standards of earlier work on *S. rosetta* in two crucial respects:

- (1) *S. rosetta* has been isolated only once (in the year 2000) from a natural sample and has never been re-observed or re-isolated since (despite extensive efforts). Thus, its natural environment remains virtually unknown, and it is impossible to correlate its diverse alternative phenotypes (observed in the laboratory) to natural environmental conditions. This contrasts with our observations on *C. flexa*, which has been reproducibly found in a well-documented environment since its discovery in 2018, and whose multicellular form consistently correlates with specific and interpretable conditions. Note that, although multicellularity and mating in *S. rosetta* can be induced in the laboratory by signals from bacteria or algae, the relevant studies have necessarily been restricted to a laboratory context: it is still unknown whether (and if so, where) *S. rosetta* naturally coexists with those bacteria or algae, whether (and where) it naturally alternates between unicellular and multicellular forms, etc.
- (2) *S. rosetta* only forms colonies clonally, which has been assumed to be the rule for all choanoflagellates. This is evident in statements in leading textbooks and review papers in the field:

- “Choanoflagellates are known to form colonies by clonal division.” (Najle & Ruiz-Trillo, The protistan origin of animal cell differentiation; in Hejnol & Leys (ed.), *Origin and evolution of metazoan cell types*, CRC Press, 2021)
- “Well-studied examples of simple clonal multicellularity are the colonial choanoflagellates, which form facultative colonies through the division of a mother cell without the daughter cells separating.” (Chipman, *Organismic Animal Biology. An evolutionary approach*, Oxford University Press, 2024)
- “Choanoflagellates (...) exhibit diverse forms of facultative group formation. (...) The multicellular structures are the result of clonal multicellularity: they arise when cells fail to separate after division and remain attached to one another.” (Davison & Michod, Phenotypic plasticity and evolutionary transitions in individuality; in Pfennig (ed.), *Phenotypic plasticity in evolution*, CRC Press, 2021)
- Finally, phylogenetic trees in multiple influential review papers depict “clonal multicellularity” as the only character state for choanoflagellates, consistent with the research literature anterior to our study. Two examples are shown below (clonal multicellularity in choanoflagellates is highlighted by a red arrow):

Sebé-Pedrós et al. The origin of Metazoa: a unicellular perspective, *Nature Reviews Genetics*, 2017

Ruiz-Trillo et al. The origin of metazoan multicellularity, *Ann. Rev. Microbiol.*, 2023

Thus, the conceptual novelty of our paper is two-fold: (1) it documents clonal-aggregative multicellularity, which is a new concept for choanoflagellates and, more broadly, for close relatives of animals; (2) it establishes *C. flexa* as the first choanoflagellate model system in which multiple cell phenotypes can be directly correlated with natural environmental conditions, thereby opening a new line of research.

Line 157 “In 10 out of 79 splash pools” – what about the 69 other pools?

In the 69 other pools, no sheets were found. This is visible in the graph **Figure 3E**, where magenta points indicate presence of sheets and turquoise points indicate absence of sheets.

We have added the following statement to make this clearer:

*“In 10 out of 79 splash pools, we found choanoflagellate sheets that we identified as *C. flexa* based on morphology, inversion behaviour, and 18S ribosomal RNA (rRNA) sequencing (Figures 3F and S7; movie S15; Tables S1-S2; Supplementary files S1-S4). *C. flexa* sheets were not observed in the other 69 splash pools.” [emphasis added]*

Note that many of the 69 sheet-free splash pools were outside the permissive salinity range for sheets (see **Figure 3E** and **Table S1**).

Line 166 “We found sheets in 14 out of 71 splash pools” - what about the 57 other pools?

Same as above. We have added the following statement:

“We observed sheets in 14 (out of 71) splash pools but not in the 57 others, with a 94 ppt salinity upper bound for sheet occurrence (Figures 3E and S6D).”

Claim 3. “Different splash pools contain different strains of *C. flexa* between which aggregation is constrained by kin recognition, a hallmark of aggregative multicellularity” Another point concerns the analysis of kin recognition. First, the kin recognition index is difficult to interpret, given the small apparent dynamic range in Fig 5H and the distributions in Fig S15B having a ‘normalised percentage’ y-axis ranging from -1 to 1. A more intuitive measure may be the (mean) absolute difference (in percentages) or other measures that preserve the percentage (or ratio) values.

Here, feedback from Reviewer 1 concurs with that of Reviewer 3, who suggested to use a more established ‘segregation index’ directly based on the relative abundance of genotypes (from Estrala & Brown *PLoS Comp. Biol.* 201, PMID: 24385891).

Considering two strains A and B, the local segregation of strain A within an individual colony i is defined as:

$$seg_A(i) = \frac{n_{A,i} - 1}{N_i - 1}$$

Where $n_{A,i}$ is the number of cells of strain A in colony i , and N_i is the total number of cells in colony i .

The global segregation of strain A in the population is simply the average of segregation values for A in individual colonies:

$$seg_A = \frac{1}{N_A} \sum_i n_{A,i} * seg_A(i)$$

Where N_A is the number of colonies in the batch (*i.e.* in the biological replicate under study). Finally, the segregation index S_A normalizes segregation by the proportion P_A of cells of strain A in the whole cell population:

$$P_A = \frac{N_A}{N_A + N_B}$$

$$S_A = \frac{\text{seg}_A - P_A}{1 - P_A}$$

If strains are initially mixed in equal proportions ($P_A = 0.5$), the segregation index can range from -1 (if strain A only aggregates with strain B and completely avoids aggregating with itself) to 1 (if strain A only aggregates with itself and completely avoids aggregating with strain B).

The resulting segregation index (S index) values are now plotted in a revised heatmap (**Figure 6G**):

And in a revised supplementary panel, confirming that the segregation index for the combination of Strain 1 (ChoPs7) and Strain 2 (M44B clone 1C5) is significantly larger than 0 (the value measured in negative control experiments where ChoPs7, stained with two different colours, is left to aggregate with itself; **Figure S22C**):

The revised analysis supports kin recognition between Strain 1 and Strain 2 and quantifies with a more readily interpretable index. Thanks for this useful suggestion.

Second, in Fig S15C, the kin-recognition index should not have a negative value on the y-axis, since variance cannot have a negative value.

We agree that the extension y-axis of the previous **Figure S15C** the negative range was unnecessary (although no negative value was plotted since, as noted by the referee, variance cannot be negative). This graph has now been replaced by one plotting the revised kin recognition index above (**Figure S22C**).

Note that the entire section needs to be proofread. Fig 5K is not described in the legend, Fig 5A is not cited in the text. In L342, it is unclear if the authors are meant to refer to S15-S16 (rather than S5-S6). Fig S15 legend refers to the wrong figures (L1177).

Thanks for catching these. These have all been fixed as requested.

This leaves us with an open question: is there a link between kin recognition and SNPs on the transmembrane proteins? Perhaps the SNPs on these transmembrane proteins can be correlated in the space?

This is a great suggestion. We have now quantified pairwise sequence differences between candidate kin recognition proteins across the three strains characterized. We did find a correlation between kin recognition index (left) and degree of amino acid substitution in transmembrane proteins (right; **Figure S23A**):

Interestingly, the domains showing the strongest signature of diversifying selection ($Ka/Ks > 2$) between Strain 1 and Strain 2 are cadherin domains, and the individual cadherin with the highest Ka/Ks shows a strong correlation between number of non-synonymous substitutions along its sequence and strength of kin recognition (**Figure 6J-J'**):

Given this evidence, we now consider this cadherin the currently most promising candidate for the kin recognition locus. However, testing this hypothesis will require dedicated functional experiments representing a whole different project which is planned for the future. Overall, we thank the reviewer for this useful suggestion.

Claim 4. “Challenging former generalizations about the choanoflagellate-animal lineage and expanding the option space for the development and evolution of multicellularity” Given the phylogenetic position of *C. flexa*, the implication of mixed clonal-aggregative multicellularity (or L435 “evolutionary interconversions”) for metazoan evolution is very speculative.

For example, in Fig S6D, the mixed clonal-aggregative multicellularity observed in *C. flexa* is likely not ancestral within choanoflagellates. Thus there is caveat that aggregative multicellularity may not have been observed thus far in other choanoflagellates. it would have been worth performing an ancestral state reconstruction on the natures of multicellularity (clonal, aggregative or mixed) and adapting the discussion to the inferred ancestral states.

While an ancestral state reconstruction would be very interesting and is indeed an important long-term perspective in the field, it would, in our opinion, be currently premature for three main reasons:

(1) The phylogeny of choanoflagellates is in a state of flux, with the positions of major branches varying significantly between different studies (compare for example: PMID 39847813, 27765632, 21895836, 18922774, 28318975).

(2) Even if the phylogeny were perfectly known, choanoflagellate life history is not understood for the vast majority of species – except *Salpingoeca rosetta* and now *C. flexa*. For most species, only a single cell phenotype has been described (sometimes unicellular, sometimes multicellular, with little correlation with the phylogeny – almost certainly reflecting the randomness of the first observed phenotype more than real life history). For example, *Salpingoeca infusionum* was cultured for years in a unicellular form, until it was serendipitously discovered it could become multicellular (at least in the laboratory). Reciprocally, *Salpingoeca rosetta* was first isolated and described as a multicellular rosette (exclusively), and it took years of painstaking laboratory work to discover it could also differentiate into other forms (such as single cells and chain colonies).

Such detailed work has simply not been done for the overwhelming majority of choanoflagellate species, making mapping character states too unreliable for ancestral state reconstruction.

(3) Finally, even when multicellular forms are described, it is almost always unknown whether their multicellularity is clonal, aggregative, or clonal-aggregative. Clonality has only been rigorously shown in a single species so far, *S. rosetta*. With our study, *C. flexa* becomes only the second choanoflagellate for which multicellular development has been rigorously characterized (which came with the surprise of clonal-aggregative multicellularity).

Thus, we would refrain from bold statements about animal origins (which would in our view be premature). Instead, we think our study provides an important conceptual advance by documenting a new and unexpected mode of choanoflagellate multicellularity and by placing it in its natural ecological context.

Consistently, we have toned down the statement quoted by the reviewer into:

“Our findings challenge former generalizations about choanoflagellates and expand the option space of choanozoan multicellularity.”

Finally, we have also removed the mention of possible interconversions.

Other points

Regarding replication, the manuscript does not provide sufficient details or data to validate the robustness of the findings. The authors state that “all experiments were replicated and quantified at least twice” and that additional unquantified replications were performed to ensure repeatability. However, this level of replication sounds insufficient, and it is unclear what was repeated and how many times. The data from replicate experiments, including the number of independent observations (number of colonies observed, number of independent samples from independent pools), should be included. All experiments should be independently replicated at least three times, with clear data supporting the conclusions drawn.

The phrase quoted did not appear in the paper, but was present in the co-submitted “Transparent reporting form”. We agree this was an overly cursory summary of our statistical methods. We have now re-written the Transparent reporting form to include more details about replicates:

“All experiments were performed in at least three replicates. Full information on the number biological replicates (independently cultured batches of cells) and of technical replicates (independent experiments with cells sub-sampled from the same biological replicate) can be found in Materials and Methods.”

In the previous version of the paper, the only experiment that had been performed in two replicates was the time-lapse of aggregation. We have now replicated this experiment two additional times, raising the number of replicates to four. All experiments have been replicated at least three times. To dissipate any doubt regarding the extent of replication in the paper, we have now listed all replicates (biological and technical) and sample size for each experiment in the Materials and Methods. This information is also summarized in the table below for the reviewer’s interest. We

hope it is now clear that all experiments have been performed in at least three replicates (and often many more). If the reviewer thinks that additional replicates are necessary for certain individual experiments, we would be happy to provide them upon request.

Exp_Number	Details	Related Figures	Biological Replicates	Technical Replicates	Total Replicates (biological *technical)	Sample size (n)
Exp0 (intro)	Micrograph of flagella-in sheet (BF)	Fig. 1B	3	1	3	n=56
Expo0 (intro)	Micrograph of flagella-out sheet (Airyscan)	Fig. 1D	7	2	13	n=30
Exp1	Single cell division	Fig. 1E-F; Movie S1	3	1	3	n=3
Exp2	Clonal expansion	Fig. 1G; Fig. S1; Movies S2-S3	2	2	4	n=4
Exp3	Aggregation dynamics (BF)	Fig. 2B-D; Movies S4-S6	4	1	4	n=4
Exp4	Timepoint Aggregates (Airyscan)	Fig. 2E-G; Fig. S2; Movies S7-S11	2	2	4	n=75 sheets
Exp5	Dual labelling timelpase (BF)	Fig. S3A; Movie S12	3	1	3	n=3
Exp6	Dual labelling (Airyscan)	Fig. 2H; Fig. S3B-C; Movie S13	2	2	4	n=16 sheets
Exp7	Aphidicolin Treatment	Fig. 2I; Fig. S4B-C	3	3	6	n>1474 particles per condition
Exp8	Aphidicolin Growth Curve	Fig. S4A	3	3	6	n=6 per condition
Exp9	Aggregation on fixed vs. Live cells	Fig. S5	3	2	6	n>2566 particles
Exp10	Curaçao Expeditions	Fig. 3D-E; Fig. 6A; Fig. S6, Movie S14	NA	1	150	n=150 splashpools
Exp11	Sheets isolated from the field (BF)	Fig. 3F; Fig. S7A-C; Movie S15	4	1	4	n=18 sheets
Exp12	Strain isolation + 18S rRNA	Fig. S7D	6	1	6	n=6
Exp13	Splash pool natural evaporation experiment	Fig. 3G-H; Fig. S8	15	1	15	n=15 splashpools
Exp14	Soil rehydration	Fig. 3I-K; Movie S16	2	1	32	n=32
Exp15	Splashpool Sheet evaporation (field)	Fig. S9	2	2	4	n=4
Exp16	Artificial gradual evaporation - quantification and imaging (BF)	Fig. 4D-E; Fig. S10B; Movie S17	11	1	11	n=11
Exp17	Control evaporation - loss of multicellularity	Fig. S10C	2	9	18	n=18
Exp18	Live imaging sheet rehydration after gradual evaporation (BF)	Fig. S10; Movie S18	2	2	4	n=4

Exp19	Artificial gradual evaporation - Cysts induction (BF)	Fig. 4E-H	3	2	6	n=6
Exp20	Addition of salts - Induction of cysts (BF)	Fig. S11A-C	2	2	4	n>4
Exp21	Addition of salts - Growth curve	Fig. S11D-E	2	2	4	n=4
Exp22	Artificial gradual evaporation - Cysts vs flagellates (Airyscan)	Fig. 4I-K; Fig. S12	3	2	6	n=12 cells
Exp23	Artificial gradual evaporation - Growth curve	Fig. 4L; Fig. S13	3	2	6	n=6
Exp24	Artificial gradual evaporation - Morphometrics comparing N:C cysts vs. Flagellates	Fig. S14	3	2	6	n=65
Exp25	Survival post rehydration cysts vs. rapidly evaporated sheets	Fig. 4N; Fig. S15A	1	6	6	n=6
Exp26	Captured bacteria single cell vs. Sheet (BF and quantification)	Fig. 4O; Fig. S15B	3	2	6	n>163 particles
Exp27	Latrunculin B Effect on sheet dissociation	Fig. S16A-D	3	2	6	n>465 sheets per condition
Exp28	Latrunculin B Effect on aggregation	Fig. S16E-H	3	2	6	n>2900 particles per condition
Exp29	Aggregation vs salinity	Fig. 5A; Fig. S17A	3	2	6	n>1832 particles per timepoint and condition
Exp30	Aggregation vs. Cell density	Fig. 5B; Fig. S17B	3	2	6	50<n<16804
Exp31	Control vs. Aggregative - Morphometrics (Airyscan)	Fig. 5D-F; Fig. S17C-J	3	2	6	n=87 sheets
Exp32	Control vs. Aggregative - inversion quantification (%)	Fig. 5G; Fig. S18	3	2	6	n=60
Exp33	Dual labelling FRAP	Movie S19	3	1	3	n=3 sheets
Exp34	Incorporation of green cells into sheets	Fig. S19	3	2	6	n=48
Exp35	Capture efficiency assay Control vs. Aggregative	Fig. 5H	3	2	6	n=6
Exp36	Cell density splashpools	Fig. 6A	1	12	12	n=12 splashpools
Exp37	WGS of strains isolated from the field	Fig. 6B-D; Fig. S20, S22	3	1	3	n=3
Exp38	Kin recognition experiment between Strain 1, 2 and 3	Fig. 6E-H; Fig. S21	2	2	4	n=530
Exp39	Kin recognition Cflexa vs Srosetta	Fig. S22	3	2	6	n=211 particles
Exp40	Bacteria density vs. Salinity (included in the response to reviewers)	NA	3	2	6	n=6

Note that the use of the term “development” throughout the manuscript is not fully justified. First, there is no “development” in these organisms (in other words, there is no ‘developmental program’).

The word “development” has been commonly used in the scientific literature to refer to the formation of choanoflagellate colonies (see PMID 20971426, 32220848, 32496191, 40695282), including by ourselves (PMID 41037400). Nonetheless, the concept of development is not central to this study, and we have thus removed this word from the revision.

Secondly, it is critical to demonstrate that what is referred to as “aggregative colonies” reflects true multicellular (albeit simple) behavior and is not the result of random cell proximity, as explained above.

We agree with this point and have addressed it above.

The difference between the terms agglomeration and aggregation is unclear, did I understand correctly that cells aggregate into sheets and then sheets can get together to form agglomerates? Is this really meaningful?? It is still unclear to me what is the ‘unit organism’, the sheets (aggregates)? or the agglomerates?

We agree the word “agglomeration” was unnecessary to our point and have removed it.

Improved visualisation. Fig S16: the values on the heatmap are not easy to see. Perhaps, the values can be rounded to 2 decimal points. Fig 5D: bootstrap values can be written directly. Fig S9D: images are pixellated. Fig 5E and 5F: scale bar needed.

We have now corrected all these as requested, except former **Figure S9D** where the resolution of the panel was limited by the raw data (these images were captured with a 5X objective). To complement this limitation, we have included the corresponding movie in the supplementary material (**Movie S18**).

Fig 2F: a control without DMSO treatment is missing. Each replicate begin with same number of cells.

We confirm that all replicates started with the same number of cells and now state this explicitly in the Materials and Methods:

“Single cells were concentrated to 3×10^5 cells/mL for each replicate, and 200 μ L were transferred into an μ -slide 8-well chamber (#80826, Ibidi).”

DMSO was included in the control as it used as a solvent for aphidicolin, making it a necessary control. However, aggregation also proceeds robustly in the absence of DMSO, as documented by all the (many) other aggregation experiments in the paper (**Figures 2B-E** and **S3** for example). Moreover, proliferation is not affected by DMSO (see **Figure S4A**).

L120: a space after the full stop is missing. L396: space needed “aggretativemulticellular”.

Thanks for catching this. This has been corrected.

Movie S4 – “Dissociated single cells reform sheets by cellular aggregation” I don’t understand what I am seeing, is this a time lapse? Looking at the individual images I have a lot of trouble in understanding what is going on. The trajectory of each cell should have been tracked independently to be able to follow their pathway and clearly show what it is supposed to show (sheet reformation?).

We understand the difficulty of visually following cells in our former supplementary movie. Tracking cells during the formation of choanoflagellate colonies is challenging and is not usually done, as choanoflagellates swim at high speed (~10-20 $\mu\text{m}/\text{sec}$) and in three dimensions. This is two orders of magnitude faster than the fastest cell migration recorded in animal embryos (the primordial germ cells of zebrafish, with a speed of 20-30 $\mu\text{m}/\text{minute}$). Tracking approaches that are common practice in animal embryology can thus be surprisingly challenging in choanoflagellates.

Nonetheless, we have now overcome these limitations by acquiring new time-lapse movies of aggregation after confining cells in a small volume (a 0.2 μL droplet) that restricts their movements in x, y and z, and imaging at high frequency (one frame per second). This has allowed us to produce two new supplementary movies in which aggregating cells can be visually followed: **Movie S5** (which shows two cells aggregating) and **Movie S6** (which shows three small sheets fusing). Note that we have manually tracked the four cells in **Movie S6** that engage in aggregation with neighbouring colonies. This allows better visualization of aggregation and of the ensuing maturation process from early irregular aggregates into mature cell monolayers. Snapshots from these movies have now been included in our main figure: **Figure 2C-D**. Thank you for this useful suggestion that clearly improved visualization of the aggregation process.

All the other ‘movies’ are 3D representations of groups of cells –are these ‘multicellular units’ ? they seem to be rather clumps of cells (see my comment above).

We have addressed this point above.

Which strain is used for the final genome assembly?

The strain used for the genome assembly is mentioned in the Materials and Methods:

“ChoPs cultures for genome sequencing and assembly were established by thawing a low-passage polyxenic, light-responsive C. flexa culture initially isolated in 2018 and reported in a previous study (Brunet et al. 2019). That culture was not isolated in Shete Boka National Park but at another site on the Curaçao coast, where C. flexa was initially discovered (12°13’38.9” N 69°00’47.0” W).”

Single-sheet-bottlenecked culture vs clones. Unclear 344-346

We have now rephrased this to clarify the difference between single-sheet-bottlenecked cultures and single-celled-bottlenecked cultures:

“Each strain was established by manual isolation of a single sheet followed by amplification in laboratory cultures (referred to as “single-sheet-bottlenecked” cultures or “SSB”). Isolation of single cells from each single-sheet-bottlenecked culture allowed establishment of clonal strains (referred to as “single-cell-bottlenecked cultures” or “clones”).”

Single-cell-bottlenecked cultures are sometimes referred to as “clones” because all cells in these cultures are expected to be genetically identical, since they are all descendants of a single isolated cell. On the other hand, given the possibility of aggregation, single-sheet-bottlenecked cultures (established from a single isolated sheets) could in principle comprise different genotypes (as discussed above). Thus, we do not refer to single-sheet-bottlenecked cultures are “clones”.

L340: What is a high confidence SNP?

We had initially referred to SNPs as “high confidence” if they passed certain standard thresholds of coverage (read depth > 4) and quality score (> 30). We have now removed this phrase from the main text as we agree it was confusing without additional explanations.

Supp. L987-988: the R version might be more informative than the RStudio version.

Thanks for catching this. We now refer to both the R version and the Rstudio version in the Materials and Methods:

“All statistical analyses were performed using the R Stats Package version 3.6.3 (R Core Team, 2020).”

Control for bacterial density (w.r.t. salinity)

We have now quantified bacterial growth under diverse salinities, and found that *H. oceanii* food bacteria can grow over the full range of salinity investigated in our laboratory experiments (from 1X to 3X seawater salinity – albeit slower at 3X).

Moreover, directly adding salt to cultures causes dissociation/encystation of *C. flexa* sheets without causing bacterial death or depletion (as shown above). Thus, changes in bacterial density are not the cause of the dissociation/encystation response of *C. flexa* to hypersalinity.

No Statistics are presented for 2F.

Following the reviewer's request, we have now added p-values (Mann-Whitney test) comparing aggregate size in DMSO- vs. Aphidicolin-treated samples (both in new **Figures 2I** (old 2F) and **S4C**). Size was not significantly different between control and aphidicolin-treated aggregates.

Non-kin aggregates: are there difference in sheet size?

This is an interesting question, as lower size could be another indication of kin recognition restricting aggregation. However, we did not observe size differences between kin and non-kin aggregates (see **Figure S22A**): $p=0.42$ by the Mann-Whitney test.

Referee #2 (Remarks to the Author):

General comments:

This is a beautiful work bridging environmental biology with modern cell and developmental biology. It combines a series of meticulously designed lab and field experiments, including advanced live-imaging, fixed-imaging microscopy, isolation, sequencing, and genomic analysis of the choanoflagellate *Choanoeca flexa*, a close relative of animals. The authors make several key discoveries: they first show, using live-imaging, that multicellular sheets of *C. flexa* form not only through clonal division but also through aggregation—a previously unreported mechanism for choanoflagellates. Field experiments, along with lab experiments that replicated natural environmental conditions, reveal that the life cycle of *C. flexa* is driven by salinity fluctuations in its splash pool habitat. As salinity rises due to evaporation, multicellular sheets dissociate into cyst-like forms; when salinity decreases, multicellularity is restored via both aggregation and clonal division. This suggests an adaptive strategy for rapid multicellular development, likely to enhance bacterial prey capture during the short-lived splash pool conditions. Additionally, the authors provide a new, fully annotated genome assembly for *C. flexa* and uncover evidence that aggregation is constrained by kin recognition, reducing potential conflicts. This elegant and thorough work deepens our understanding of the flexible, environment-driven mechanisms that may have shaped the evolution of multicellularity.

In principle, this is a beautifully executed study with precise experiments. However, I offer a few points below that could help strengthen the authors' interpretations and conclusions.

Thanks a lot for your positive comments, and for your constructive criticism that significantly improved the manuscript.

Major points :

1. Structural flow: The current structure of the paper feels like two distinct stories: one focused

on the aggregation process and another centred on the life cycle of *C. flexa* in its dynamic splash pool environment. While each component is well-explored, the connection between them is limited, which makes the overall narrative somewhat less cohesive. A potential approach could be to reframe the story around the entire life cycle of *C. flexa*. This would start by describing the environmental dynamics of splash pools, emphasizing the role of salinity fluctuations in driving the organism's life stages, from desiccation and cyst formation to the rapid re-emergence of multicellularity. By highlighting that multicellularity reappears faster than clonal division alone could account for, the narrative could naturally introduce the role of aggregation as a key mechanism for rapid colony formation. This structure would make it easier to integrate the data on aggregation and kin recognition, showcasing them as adaptive responses shaped by the fluctuating environment.

Alternatively, if the current structure is maintained, it would be beneficial to strengthen the textual links between the environmental life cycle data and the multicellular behavior. Explicitly connecting the changes in salinity to the shifts in multicellularity throughout the text would provide a more cohesive and integrated interpretation of the findings.

Thanks for this point. While we really see the paper as a single cohesive story, we agree it could have unintentionally come across as two distinct studies. We have extensively revised the manuscript to make the connection between clonal-aggregative multicellularity and the splash pool environment more explicit. As is evident from the new version, we have taken the second route you suggest: without drastically rearranging the original structure, we added novel paragraphs, figures and experiments that more directly connect clonal-aggregative multicellularity to variations in salinity.

To better bridge the two parts of the study and directly link the life cycle dynamics with multicellular behavior, I recommend additional experiments focused on the speed of sheet development under different stress conditions: Measure the speed of multicellular sheet formation in cultures that have been pre-stressed by high salinity versus those that have not been salinity-stressed. This would involve dissociating all cultures into single cells, with one group exposed to prior salt stress and the other group left unstressed.

Assess the rate of sheet formation both in isolated (stressed or unstressed) and mixed conditions (stressed cells combined with unstressed cells) after switching to conditions favouring division and sheet formation. Time-lapse imaging could be used to quantify the speed of colony formation in each case. In parallel, measure the cell division rates under these conditions over time (as some division rate data may already be present in the study) to directly compare the contribution of clonal division versus aggregation in sheet formation. These experiments would help clarify the link between the life cycle, particularly salinity stress, and the "driving" force behind multicellular sheet formation.

Thanks for this great suggestion. We have designed an experimental pipeline to directly compare the contribution of clonal division versus aggregation in sheet formation, as you suggest. This converges with one of Reviewer 1's main point, asking to more directly test the selective advantage of clonal-aggregative multicellularity as a plastic and versatile strategy to deal with a broader range of conditions.

We investigated the effect of two relevant environmental parameters that we had found to vary between splash pools: salinity and density of *C. flexa* cells (**Figure 5A-B** and **S17A,C**). We found that medium-high salinity (~70 ppt, or about 2X seawater salinity) inhibits proliferation and thus clonality, but leaves aggregation unaffected. By contrast, and unsurprisingly, aggregation is inefficient at very low cell density (at which clonality still operates). We conclude that clonal-aggregative multicellularity serves as a versatile strategy in the variable environment of splash pools.

I know some experiments are already within the paper, they however seem scattered and somewhat not cohesively presented.

2. Aggregation and orientation:

The authors present compelling evidence for aggregation through several movies and figures, showing cell aggregation at the periphery of developing sheets. They strengthen this claim with experiments using the division inhibitor aphidicolin, demonstrating that aggregation can occur independently of cell division. Interestingly, the data suggest that aggregation initially happens in a disordered manner, followed by a reorientation phase where the cells align consistently. Additionally, the use of three genetically distinct *C. flexa* strains highlights the role of kin recognition in aggregation, suggesting a non-random, selective process. While the data are convincing, I still have several questions and suggestions for further exploration:

- Could part of the observed aggregation be random, where cells become "entrapped" together initially, only to divide and reorient later? This is an open question (and one I'm unsure about or whether there is a clear way to showcase it).

To explore this, one experiment could involve adding dissociated, fluorescently labeled single cells of one color onto pre-formed colonies of a different color. Comparing this scenario (few colonies with many single cells) to a control where only single cells of distinct colors are mixed could help differentiate random aggregation from more directed, kin-based aggregation?

We have addressed this in three series of experiments:

- 1) We have now shown that fixed cells do not aggregate (even if encounters are forced by orbital agitation). This shows that aggregation is an active process, requiring living cells (new **Figure S5**).

- 2) We have shown that reorientation is independent from cell division by quantifying the morphology of sheets (including their cell orientation) resulting from pure aggregation (under medium-high salinity conditions, that inhibit proliferation; **Figure 5D-F** and **S17D-J**):

- 3) Finally, we attempted the experiment you suggested: adding fluorescently labelled cells to pre-formed non-labelled sheets (**Figure S18**). We performed this at 2X salinity, precluding cell division. We found that single cells do join pre-formed sheets and, after an initial “contact phase” of variable collision/orientation, eventually adopt a “polarized” orientation. This reinforces this point that reorientation is independent from cell division in a different biological context than the experiment above (single cells joining a pre-existing colony, rather than forming colonies *de novo*):

- The collar complex likely plays a central role in the aggregation process. A potential experiment could involve testing the aggregation behavior of single cells that lack a functional collar complex. This could be achieved by pre-treating cells with actin inhibitors or exposing them to high salinity (as suggested earlier), then mixing these non-collar cells with pre-formed colonies of a distinct color. Monitoring whether aggregation occurs and in what proportion could provide insights into the role of the collar complex in the initial cell-cell adhesion process.

Thanks for this great suggestion. To assess the role of the collar complex in aggregation, we treated colonies with the actin polymerization inhibitor latrunculin B, which caused collar retraction (**Figure S16A-D**). The effect was fast and dramatic: colonies fully dissociated within minutes and the resulting single cells failed to reaggregate (importantly, they kept swimming, indicating they were still alive).

Cells that had been dissociated first and then treated second with latrunculin B similarly failed to re-aggregate (**Figure S16E-H**):

We conclude that the collar is necessary for both aggregation and maintenance of multicellularity. This is in good agreement with microscopy observations (both electron microscopy and immunofluorescence) that show cells in *C. flexa* colonies linked by direct collar-collar contacts to the exclusion of any other visible connections (unlike in other choanoflagellates, where a shared extracellular matrix or intercellular bridges often support multicellularity).

- Just thinking out loud here, but would you expect that aggregation per se has different properties than clonal division? Would an experiment where cells only formed through aggregation (Amphidicolin treated) can be subjected to precise small sonication, vs clonal-aggregative together? Basically, trying to see whether there maybe another advantage here than speed? but some sort of biomechanical reinforcement? (again, this maybe beyond the scope).

This is an interesting idea. As outlined above, we have carefully quantified morphology and behaviour in control and aggregative colonies without finding significant differences. Thus, we would not necessarily expect biomechanical differences, although this remains to be tested.

- The reorientation of cells within the sheet is a particularly interesting phenomenon. It raises the question: in the aphidicolin experiments (where cell division is inhibited), are the aggregated cells well oriented or not?

If the cells achieve proper orientation even without division, this suggests that reorientation is driven by mechanisms independent of cell division. If not, it would imply that while aggregation can occur without division, reorientation may still depend on subsequent divisions. It would be helpful if the authors included 3D images of the sheets from the aphidicolin experiments (I checked the supplementary materials but did not find detailed imaging of cell orientation in these conditions). Moreover, in the case where cells do not fully re-orient when only forming aggregate, it would be interesting to assess the flows ? and or efficiency of feeding ? (but I understand it maybe beyond the scope of this study).

As detailed in the response to Reviewer 1, we have now addressed this point, and shown that colonies formed purely by aggregation have comparable (1) cell orientation and (2) prey capture efficiency as control sheets.

- I would encourage the authors to discuss these observations as two distinct but connected processes: (1) initial aggregation and (2) subsequent cell reorientation. It would be valuable to speculate on the possible signaling mechanisms involved in each stage. Given the known importance of calcium signaling, an experiment using EDTA/EGTA (to chelate calcium ions) could help determine whether aggregation and/or reorientation are dependent on calcium-mediated interactions for instance (although I can understand that it may be beyond the scope of this study). I just think it would be very interesting to find a condition where cells aggregate, but are unable to fully re-orient.

We share the reviewer's intuition that initial aggregation and subsequent reorientation are two distinct processes. Indeed, certain experimental results support this model. One piece of evidence is that aggregation and re-orientation operate at different timescales: initial aggregation, resulting in random and variable aggregates, occurs within seconds to minutes, while reorientation to establish a regular sheet morphology requires more than an hour. This is evident in our new live imaging time-lapse data, where we imaged both early random aggregation and later reorientation of cells to form a polarized monolayer (**Figure 2D** and **Movie S6**).

Left: pre-aggregation. Middle: early irregular aggregate. Right: polarized monolayer resulting from maturation.

Thus, it seems plausible that initial adhesion and subsequent reorientation might depend on distinct molecular mechanisms. Unfortunately, we are not aware of any pharmacological treatment that would only affect reorientation: we have observed in our lab that both actin depolymerization and calcium depletion just completely prevent adhesion and aggregation (and are even sufficient to induce dissociation of pre-existing colonies). We suspect that specifically preventing re-orientation will require more specific assays, and most probably functional genetic tools. Thus, we think this intriguing question will have to await technological advances.

The authors use three distinct *C. flexa* strains to demonstrate that aggregation may be constrained by kin recognition, limiting aggregation to genetically similar cells. While this is a valuable finding, it would be strengthened by the inclusion of an outgroup, such as *S. rosetta*. Adding *S. rosetta* cells to the aggregation assay could help determine whether aggregation is driven purely by kin recognition or if some degree of random entrainment occurs. If *S. rosetta* cells do not aggregate with *C. flexa*, it would support the idea that kin recognition plays a crucial role. Conversely, if they become entrapped, it might indicate a more passive, non-selective aggregation mechanism at play.

We thank the reviewer for this great suggestion and have now performed this experiment. The result was unequivocal: *S. rosetta* entirely failed to aggregate with *C. flexa*. This shows that aggregation is species-specific and argues against accidental “entrapment” by the prey-capturing collar as a significant contributor to aggregation, except perhaps during a very brief phase of initial contact. The results are presented in the new **Figure S20** (copied below):

A**B**
Minor

points:

- This may be a personal perspective, but I find that the beauty of this study lies in its standalone contribution, without necessarily invoking the evolutionary history of animal multicellularity. The authors have done an excellent job throughout most of the manuscript in avoiding overemphasis on the origins question, keeping the focus on the unique ecological and cellular biology aspects of *C. flexa*. However, I did notice one instance in the introduction where this theme emerges: "As in animals, multicellular development is clonal in all choanoflagellate species investigated thus far, which contrasts with the occurrence of aggregative multicellularity in more distantly related lineages, such as filastereans and dictyostelid amoebae. These observations, together with the clonal nature of animal embryogenesis, have inspired the hypothesis that animals

evolved from organisms with a simple form of clonal multicellularity, akin to that observed in certain modern choanoflagellates and in other close relatives of animals such as ichthyosporeans."

While this statement is accurate, it indirectly suggests that the authors' discovery of clonal-aggregative multicellularity might challenge the hypothesis that animal ancestors had only a simple clonal multicellular organization. The results of this study, in my view, imply that the observed clonal-aggregative behavior in *C. flexa* may be an adaptation specific to its environmental context, rather than a broader ancestral trait of choanoflagellates. The absence of such behavior in other choanoflagellate species to date suggests it might be a specialized adaptation, evolved in response to environmental pressures rather than a retained ancestral capability. However, given the prevalence of aggregation across various protist lineages, we cannot completely rule out the possibility that early ancestors had both clonal and aggregative capacities.

I realize that this is a nuanced and inconclusive point, and the true answer likely lies somewhere in between. The way the introductory sentences are framed either necessitates a more thorough discussion of both scenarios later on or could benefit from a slight rephrasing to downplay the evolutionary question upfront. This would allow the focus to remain on the unique eco-evo-cell biology story of *C. flexa*, which in itself is a remarkable narrative. Of course, I acknowledge that this is a subjective suggestion.

We agree with this point and have indeed refrained from defending a particular scenario of animal origins in this paper. We also agree that the significance of our study lies elsewhere: in the fact that it changes the way we see choanoflagellate multicellularity, and in how it places it in its ecological context. We have thus followed your suggestion and toned down the quoted part in the Introduction, which now reads:

"The best-characterized choanoflagellate, Salpingoeca rosetta, only becomes multicellular clonally²⁶, and clonal multicellularity has classically been assumed to be a general feature of choanoflagellates^{4,25,27}. However, this remains to be tested across choanoflagellate diversity. Interestingly, while animal multicellularity is purely clonal, other close relatives of animals besides choanoflagellates show diverse forms of multicellularity, including aggregation in filastereans²⁸⁻³¹ as well as cellularization of multinucleated cells^{32,33} and cleavage-like serial cell divisions in ichthyosporeans³³⁻³⁶.

Figure

comments:

- In general, figures are very beautiful and well built. One comment would be to normalize the colour palettes for the collar. In some images its in blue, in others in green and in sketches it's in green. I understand that sometimes phalloidin is used in the 405 and others in the 488 but I think normalizing these across the videos and figures (maybe grey ?) would help readers to better follow through without changing anything of the narrative.

We have fixed this, and phalloidin is now white in all figure panels.

- Figure 5, I wonder whether the red and green in J drawings of the domain are colour-blind compatible?

Thanks for bringing this up – fixed as requested.

Referee #2 (Remarks on code availability):

Not my expertise

Referee #3 (Remarks to the Author):

This elegant paper from the Brunet lab reveals an unexpected mode of multicellular development in *Choanoeca flexa*, one of the closest living relatives of animals. The authors demonstrate that *C. flexa* can form multicellular sheets through both clonal division and cellular aggregation, and show this developmental plasticity is environmentally regulated by natural salinity cycles in splash pools, the organism's native habitat.

The authors establish that *C. flexa* can form multicellular sheets through pure clonal development (cell division with retained adhesion), pure aggregation of individual cells, or a combination of both mechanisms. In its natural splash pool habitat, *C. flexa* alternates between multicellular sheets at lower salinities and unicellular cysts at high salinities or during desiccation, a cycle driven by natural evaporation-refilling dynamics. Importantly, the multicellular form shows enhanced bacterial prey capture compared to single cells, providing a compelling argument for the benefits of multicellularity (this is in addition to the morphological transitions between motile 'cup' shaped colonies that flatten into a feeding mode once they are in the light, which I hypothesize also allows them to better find prey). Different splash pool populations show genetic divergence and kin recognition during aggregation, with polymorphic cell surface proteins potentially mediating strain recognition.

This is an exceptionally rich, complete story. The technical execution of this work is outstanding. The authors employ an impressive array of complementary approaches, including field ecology and environmental monitoring, time-lapse microscopy and cellular imaging, genomics and phylogenetic analysis, behavioral assays, and quantitative analysis of cellular interactions. The scholarship is exceptional, with the work thoughtfully positioned within the broader context of multicellularity evolution, development, and ecology. The authors carefully build their argument through a logical progression of experiments, each providing multiple lines of evidence for their conclusions. This is, simply put, an stand-out paper. The integration of lab and field approaches is especially noteworthy, given how hard it can be to understand the context in which the behaviors of extant organisms likely evolved.

I enthusiastically support publication of this paper, largely as is. I have few suggestions for

improvement, and my suggestions are truly minor. The authors should not feel obligated to do any of these things simply to appease me: only do them if you think they make the paper better.

Many thanks for your positive and enthusiastic comments.

First, the "kin recognition index" could be more intuitively presented using established methods for measuring preferential assortment. The segregation index described by Estrela and Brown (2013, PLoS Computational Biology) provides a more interpretable measure, scaled between -1 and 1, where 0 represents random mixing and 1 represents pure kin recognition. This would be more accessible to most readers than the current variance-based approach, where contextualizing the measurements is challenging. It's also a more useful metric, as assortment is equivalent to 'relatedness' in inclusive fitness calculations, as it's a weighted mean scalar of preferential interactions among clonemates. It's a nice way to analyze this data.

Thanks for this very useful suggestion, which we have followed. Our revised graphs using the metric you suggested are pasted below – see the revised **Figure 6G**:

And **Figure S22C**:

Second, the paper's framing of clonal development and aggregation as mutually exclusive developmental modes might be a touch too strong. While the authors are right that these are often *presented* as a dichotomy, mixed strategies are fairly common - yeast can flocculate while also forming clonal groups, Chlamydomonas does both, some sponges, plants and fungi can fuse while growing clonally, bacterial biofilms often involve mixed strategies of clonal and aggregative development, etc. What's particularly interesting is not that *C. flexa* is the first organism shown to do both, but a) the mechanisms seem to be more tightly regulated as part of their life cycle than what you see in the examples I cited above, and b) this opens the door to aggregation playing a role in the origin of animals, which nobody has really spent much time thinking about.

We agree that there have been previously published examples of mixed clonal-aggregative strategies. In our view, the novelty of our study does not lie in the discovery of clonal-aggregative multicellularity per se, but rather in: (1) its existence among the closest relatives of animals, which contrasts with prevailing textbook generalities on choanoflagellates; (2) the fact that it is deployed as a specific life history strategy, as pointed out by the reviewer; (3) the fact that clonal-aggregative multicellularity results in colonies with extensive multicellular-level adaptations, including controlled shape and collective behaviour – features that are usually associated with pure clonal development.

We have accordingly rephrased our abstract and discussion. From the revised abstract:

“Clonal and aggregative development are traditionally considered mutually exclusive^{1-6,9}, with rare exceptions¹⁰, and evolutionary hypotheses have addressed why multicellular development might diverge toward one or the other extreme^{3,4}.” [emphasis added]

From the revised discussion:

*“The mixed clonal-aggregative multicellularity of *C. flexa* was a surprise, given that clonal and aggregative multicellularity are often depicted as mutually exclusive in holozoans^{2,79} and eukaryotes in general⁸⁰⁻⁸². However, although clonal-aggregative multicellularity had not yet been reported in close relatives of animals or in the context of a regulated life cycle to our knowledge, clonality and aggregation cooperate in the formation of diverse other structures, including bacterial biofilms⁸³, experimentally evolved clusters of bacteria⁸⁴, predator-induced groups of freshwater algae (which can even combine different species)¹⁰, clusters of budding yeast⁸⁵, and certain syncytial amoebae¹⁹. Although these diverse processes might not all represent bona fide multicellularity⁸⁶ (as the resulting structures often seem to lack multicellular-level adaptations such as controlled shape, size, or collective behaviour), they show that aggregation and cell division without separation of sister-cells can coexist. Clonal-aggregative multicellularity may therefore be more widespread than currently appreciated.”* [emphasis added]

Third, the adaptive significance of kin recognition warrants deeper exploration beyond the standard explanation of cheater prevention. I personally think people worry too much about the threat posed by cheaters. In many simple, early multicellular organisms, cheating may not be a major selective pressure. If there isn't an obvious common good within *C. flexa* colonies that would tempt cheaters, alternative explanations for kin recognition should be considered. One compelling possibility is that the benefit lies in generating genetically-structured groups - not to prevent evolutionary deterioration through cheating, but to enable evolutionary construction and the origin of new multicellular innovations.

As discussed below, we fully agree with the reviewer's comment, and have revised the text accordingly. However, we would like to take the opportunity to point out that, in our opinion, it is common goods within *C. flexa* colonies are not unlikely to exist. The most plausible one is the collective feeding flow, that benefits all cells but requires investment from each. One possible 'cheating strategy' would consist in cells that do not beat their flagellum when part of a colony, thus saving energy while exploiting flow from their neighbours. A similar potential cheater phenotype would be cells that would not actively contribute to the collective inversion behaviour (which is likely another common good: presumably, collective inversion still occurs even if not all cells within a sheet contribute, as long as a critical number of them still do).

This works through a fundamental mechanism: in the absence of assortment, selection acting on the traits of groups does not result in evolutionary change (no change in allele frequencies over generations). However, when genotypes are positively assorted within groups, as they are in this case due to kin recognition, there exists a covariance between the genotypes of individuals within

groups and selection acting on group traits. At an extreme, when groups are clonal, then selection acting on the traits of groups is acting on the genes within the cells of those groups that created the group-level trait under selection. While clonality is obviously the best for this, any positive assortment creates a statistical linkage between the traits of groups and the fitness of alleles within the group that underpin the expression of multicellular traits.

C. flexa has evolved sophisticated multicellular behaviors (i.e., cells responding to light by changing their collar angle, that changes the shape of the colony from a flat sheet to a motile cup) where there is a clear linkage between the traits of cells and the traits of groups. Selection acting on the behaviors of groups would be much more efficacious if those groups were highly assorted, as opposed to random grabs of all the different genotypes that exist in a splash pool. Thus, kin recognition might be more about enabling the evolution of coordinated multicellular behaviors than preventing exploitation. If this argument does not make sense to you, no worries, reach out to Dr. Gee and ask him for my name. He has my permission to waive anonymity, and we can chat about it. I think a quick zoom session with a virtual white board would make it super clear.

We thank you for bringing this up. We agree that kin recognition should logically facilitate the evolution of coordinated multicellular behaviour, and that this might indeed be the main selective pressure for its evolution and maintenance (rather than protection against cheaters). This idea was notably defended by Wolpert and Szathmari in their 2002 paper “Evolution and the egg” (where “egg” should be understood as “single-cell bottleneck” ensuring genetic homogeneity):

*“One mechanism for pattern formation is based on positional information: cells acquire a positional identity that is then converted into one of a variety of cellular behaviours, such as differentiating into specific cell types or undergoing a change in shape and so exerting the forces required for the formation of different structures. This and other patterning processes require signalling between and within cells, leading ultimately to gene activation or inactivation. **Such a process can lead to reliable patterns of cell activities only if all the cells have the same set of genes and obey the same rules. During evolution, it is the change in genes that leads to the creation of new patterns of development.***

*It is only through a coherent developmental programme, with all cells possessing the same genes, that organisms can evolve, and this requires an egg. **There are multicellular organisms, such as the cellular slime moulds, that develop by aggregation and not from an egg, but their patterns of cell behaviour have remained very simple for hundreds of millions of years. The evolution of more complex organisms increases the pressure to use an egg as a propagule.***”

More recently, this concept was dubbed the ‘coordination hypothesis’ and supported quantitatively by the Queller & Strassman lab in *Dictyostelium* (Jahan et al. 2025; doi: 10.1093/evlett/qrae063).

We have now added a reference to this hypothesis and cite these two papers in our discussion, making sure to not overstate the risk of cheaters, and to give at least equal weight to the coordination hypothesis:

*“Aggregative multicellularity comes with a well-known evolutionary challenge: **one aggregate can combine cells of different ancestries and potentially different genotypes, thus raising the possibility of genetic conflict**^{93–95} and, **perhaps even more importantly, limiting the possibility of genotype coevolution for the coordination of complex behaviours**^{73,96}. Nonetheless, the risk posed by chimerism can be mitigated if aggregation is restricted to close relatives – either actively by kin recognition mechanisms, or passively by a spatially structured environment limiting dispersal^{97,98}.”*
[emphasis added]

We hope this additional discussion does justice to the reviewer’s point, and would be happy to extend it or cite further references if necessary. Overall we agree that, given the importance of collective behaviour for the biology of *C. flexa*, cell-cell coordination is an extremely plausible (and possibly the main) selective pressure for the evolution and maintenance of kin recognition in this species. Thanks again for bringing this up.

Bottom line: this is a truly beautiful, elegant paper, one of the best yet from a rising star in the field of multicellularity. It’s a complete story leveraging many diverse tools of modern biology that should leave everyone happy- if you’re a field biologist, you’ll be envious of the genomic/lab data that can be brought to bear, and if you’re a lab scientist, you’ll be envious of such a clean field component that provides so much power for understanding the historical context of selection.

Many thanks again for your positive comments.

We thank all reviewers for their insightful feedback that significantly improved the manuscript.

Referee #1 (Remarks to the Author):

The authors have made an excellent job in answering the comments, I have no further concerns, congratulations for the very nice work.

Thanks for your very useful feedback and your positive appraisal.

Referee #2 (Remarks to the Author):

The authors have done an exceptional job addressing all my previous comments, and the paper is now far stronger than before. They've clearly put in a huge amount of work (actually quite a spectacular amount of work), carefully tackling every point raised, including some that were, frankly, unreasonable from Reviewer 1.

This is a game changing piece of work for how we think about choanoflagellate multicellularity. The data quality, depth, and clarity of the story are remarkable. I also want to emphasize that the request for an additional replicate, one that would have required another field trip to Curaçao, was unnecessary and unrealistic, and I fully support the authors decision not to pursue it. With the new data testing only aggregative sheets vs clonal ones, with all the new image datasets, sequencing, quantifications, and just a new written manuscript, this has over exceeded my expectation for a revision.

Overall, this is a truly important and field defining study. I'm fully and enthusiastically in favor of publication in its current form, and would hardly understand how such work can't make it through.

Thanks for your very useful feedback and your positive appraisal.

Referee #3 (Remarks to the Author):

I am very happy with these revisions (both for my own comments and those of the other referees). I'd like to restate that I think that this paper is very deserving of being published in Nature: not only are the conceptual issues critical for understanding animal origins (i.e., understanding the life cycles of animal relatives, which may inform our understanding of animal ancestors, is critical for understanding how simple groups of cells evolved into functionally-integrated organisms), but the experiments and scholarship are top notch. Vanishingly few papers combine fieldwork and field-leading cell biology, and knowing the environment these organisms exist in, and thus likely evolved in,

makes the cellular mechanisms all the more powerful. Fun fact: *S. rosetta*, the choanoflagellate that Nicole King brought to fame, has been isolated from the ocean exactly one time, and has never been seen again. Essentially all evolutionary hypothesis about this organisms remain ungrounded in ecological reality, and yet it's still been profoundly impactful for our understanding of animal multicellularity. The Brunet lab is doing some of the most elegant work in the field- with this paper being among their finest work to date.

Thanks for your very useful feedback and your positive appraisal.